# Predicting Post-Hepatectomy Liver Failure in HCC Patients: A Review of Liver Function Assessment Based on Laboratory Tests Scores

**DOI:** 10.3390/medicina59061099

**Published:** 2023-06-07

**Authors:** Alessio Morandi, Matteo Risaliti, Michele Montori, Simone Buccianti, Ilenia Bartolini, Luca Moraldi

**Affiliations:** 1HPB Surgery Unit, Department of Experimental and Clinical Medicine, Azienda Ospedaliero Universitaria Careggi, 50134 Florence, Italy; matteorisaliti@gmail.com (M.R.); simone.buccianti@unifi.it (S.B.); ilenia.bartolini@gmail.com (I.B.); 2Clinic of Gastroenterology, Hepatology, and Emergency Digestive Endoscopy, Università Politecnica delle Marche, 60126 Ancona, Italy; m.montori@pm.univpm.it

**Keywords:** hepatocellular carcinoma, HCC, PHLF, Child–Pugh, MELD, ALBI, APRI, PALBI, FIB-4

## Abstract

The assessment of liver function is crucial in predicting the risk of post-hepatectomy liver failure (PHLF) in patients undergoing liver resection, especially in cases of hepatocellular carcinoma (HCC) which is often associated with cirrhosis. There are currently no standardized criteria for predicting the risk of PHLF. Blood tests are often the first- and least invasive expensive method for assessing hepatic function. The Child–Pugh score (CP score) and the Model for End Stage Liver Disease (MELD) score are widely used tools for predicting PHLF, but they have some limitations. The CP score does not consider renal function, and the evaluation of ascites and encephalopathy is subjective. The MELD score can accurately predict outcomes in cirrhotic patients, but its predictive capabilities diminish in non-cirrhotic patients. The albumin–bilirubin score (ALBI) is based on serum bilirubin and albumin levels and allows the most accurate prediction of PHLF for HCC patients. However, this score does not consider liver cirrhosis or portal hypertension. To overcome this limitation, researchers suggest combining the ALBI score with platelet count, a surrogate marker of portal hypertension, into the platelet–albumin–bilirubin (PALBI) grade. Non-invasive markers of fibrosis, such as FIB-4 and APRI, are also available for predicting PHLF but they focus only on cirrhosis related aspects and are potentially incomplete in assessing the global liver function. To improve the predictive power of the PHLF of these models, it has been proposed to combine them into a new score, such as the ALBI-APRI score. In conclusion, blood test scores may be combined to achieve a better predictive value of PHLF. However, even if combined, they may not be sufficient to evaluate liver function and to predict PHLF; thus, the inclusion of dynamic and imaging tests such as liver volumetry and ICG r15 may be helpful to potentially improve the predictive capacity of these models.

## 1. Introduction

Outcomes of liver resection for hepatocellular carcinoma (HCC) have improved in recent decades due to the introduction of minimally invasive techniques, new parenchymal transection devices, enhanced critical care and a better selection of patients eligible for resective surgery. Despite advances in liver surgery, posthepatectomy liver failure (PHLF) is still one of the most feared complications after liver resection, mainly in patients undergoing resection for HCC. In fact, HCC occurrence is frequently associated, up to 80% of cases, with fibrotic or cirrhotic liver disease, resulting in impaired hepatic function and a consequent increased risk of PHLF [1].

PHLF is characterized by fluid retention, coagulopathy, non obstructive jaundice and increased susceptibility to infections. This complication remains an important cause of perioperative morbidity and mortality in HCC surgery, resulting in prolonged hospital stay, increased health care costs and poor long-term survival [2].

For these reasons, we need to predict the risk of PHLF before performing a liver resection in patients with HCC, especially when it is associated with cirrhosis, in order to select patients who are candidates for resective surgery and to exclude others who can be treated by other modalities, such as transplantation, transarterial chemo-embolization and thermo-ablation [3].

Although several methods are available to assess preoperative liver function, there are currently no standardized criteria for predicting the risk of PHLF in HCC patients. In this context blood tests are often the first and least invasive as well as the least expensive method of assessing liver function.

By analysing the most recent and relevant literature, this narrative review aims to investigate recent advances in the assessment of perioperative liver function based on laboratory tests scores in patients with HCC.

## 2. Definition of PHLF

Posthepatectomy liver failure (PHLF) is one of the most severe complications after major hepatectomy and the leading reason for post-operative short-term mortality [4]. The lack of a universal definition for PHLF in the past has led to a wide variation in the reported incidence of PHLF in the literature. Typically, it is estimated to be between 8% to 12% [5]. However, some studies have reported significantly higher rates of occurrence, with PHLF rates as high as 48.5% [6].

Various definitions of PHLF have been proposed. Many of these are based on specific laboratory test or challenging formula that may limit their daily clinical use [7] (Table 1). Balzan et al. in 2005 were the first to systematically describe the PHLF, basing their definition on blood samples (bilirubin and prothrombin time) on postoperative day 5 [8]. Later, Mullen and colleagues in 2010 introduced another definition based only on peak bilirubin greater than 7 mg/dL [9]. The main limitation of these definitions are related to their binarity, they can distinguish only between PHLF and non-PHLF. To solve this problem the International Study Group of Liver Surgery (ISGLS) proposed a standard definition and a grading system of PHLF in 2011, defining it as postoperatively acquired deterioration in the ability of the liver to maintain its synthetic, excretory and detoxifying functions, which are characterized by an increased international normalized ratio (INR) and concomitant hyperbilirubinemia on or after postoperative day 5. The spectrum of PHLF severity can vary, ranging from a mild reduction in liver function that does not require any particular treatment (grade A) to a deviation from the regular clinical course, manageable without invasive treatment (grade B) or requiring invasive treatment (grade C) [10]. The mortality rate for PHLF patients increases significantly depending on the grade, with rates for A, B and C reported to be 0%, 12% and 54%, respectively [11,12]. This system is considered simple, easy to apply and well standardized, allowing easy comparison between different centers [12,13].

## 3. Liver Function Assessment Based on Laboratory Tests Scores

### 3.1. Child–Pugh Score

The Child–Pugh Score (CP score) was originally described in 1964 by Child and Turcotte and then modified by Pugh et al. as a tool to select patients who would benefit from portosystemic shunt surgery. This score is based on five laboratory and clinical parameters such as ascites, hepatic encephalopathy, total bilirubin, albumin and prothrombin time or international normalized ratio (INR) [14,15]. For each parameter is given a score from 1 to 3, with 3 representing the most severe impairment. The final score is calculated from the sum of these points, which range from 5 to 15, and allows patients to be divided into three classes: A—good hepatic function (score 5–6), B—moderately impaired hepatic function (score 7–9) and C—advanced hepatic dysfunction (score 10–15) (Table 2). Typically, patients with class A cirrhosis may be suitable candidates for hepatectomy, those with class B cirrhosis should be approached with caution, while those with class C cirrhosis should generally avoid liver resection.

Although the CP score has traditionally been used to assess the severity of hepatic dysfunction in cirrhosis and for staging and management patients of with HCC, it has some limitations, such as it does not consider renal function, the evaluation of ascites and encephalopathy is subjective and may vary according to physician judgement, some indexes such as ascites and serum albumin are interrelated and all five parameters have equal weight [16]. In addition, when the CP score is used in patients without liver cirrhosis, as in some cases of patients with HCC, it typically remains within normal limits, even in the presence of significant underlying liver dysfunction. Therefore, in HCC patients, CP score may not provide accurate discrimination among patients, and many of these patients are classified into grade A [17]. However, the use of CP score in the preoperative setting is still strongly recommended by Eastern and Western guidelines and remains the most widely use in clinical practice to estimate the extension of resection that can be tolerated in cirrhotic livers [3,4,5,6,7,8,9,10,11,12,13,14,15,16,17,18].

### 3.2. MELD Score

The Model for End Stage Liver Disease (MELD) is a quantitative assessment tool derived from a mathematical equation that incorporates three objective serum measurements: international normalized ratio (INR), serum creatinine and serum bilirubin (Table 3). Originally was designed to predict outcomes for patients undergoing elective transjugular intrahepatic portosystemic shunts [19]. Subsequently, this scoring system has been widely used as an index of severity disease in cirrhotic patients and has been adopted as the standard approach for prioritising patients on liver transplant waiting list in many countries around the world [20]. Moreover, the MELD score has been demonstrated to be useful for predicting surgical risk, outcomes and non-transplant surgical mortality in cirrhotic patients [21]. The main benefits of MELD score are that its variables are objective and not easily affected by external factors, and for each one it is assigned a weight based on its impact on the prognosis. However, the need for computation may make it less practical at the bedside, and the Child–Pugh score remains easier to use in this setting [22].

In cirrhotic patients eligible for HCC resective surgery, this score has demonstrated its utility as a reliable predictor of postoperative morbidity, PHLF and survival outcomes [23]. However, its capacity to predict outcomes decrease in non-cirrhotic patients undergoing liver resection for HCC [24]. Cucchetti et al. observed that patients with a MELD score greater than 11 had a significantly higher incidence of PHLF (37.5%) after hepatectomy, resulting in a longer hospital stay and a shorter 1-year survival compared to patients with a lower MELD score. In contrast, no cases of postoperative liver failure were reported for patients with a MELD score below 9 [25]. Similarly, Teh et al. found an increased risk of perioperative mortality in patients with a MELD score greater than 9 undergoing liver resection for HCC, and recommended that hepatic resection (minor or major) should only be performed in patients with a score below or equal to 8 to minimize the risk of complications [23]. Combining MELD score with the CP score may provide a more effective way to stratify patients according to their liver function. For example, among patients with CP grade A who are theoretically eligible for surgery, MELD evaluation can help to identify those who have a high risk of developing PHLF and should therefore avoid liver resection [23].

### 3.3. ALBI and Related Scores

The albumin–bilirubin (ALBI) score was introduced by Johnson et al. in 2015 as an alternative tool for assessing liver function in patients with HCC. This score is based only on serum bilirubin and albumin level and eliminates subjective parameters such as encephalopathy and ascites included in CP score. It is calculated using a mathematical formula: ALBI score = 0.66 × log_10_ (total bilirubin [μmol/L]) − 0.085 × (albumin [g/L]) (Table 3). Similar to the CP score, ALBI score stratified patients into three categories: grade 1 (≤−2.60), grade 2 (>−2.60 to ≤−1.39) or grade 3 (>−1.39) [17]. There is a direct correlation between the increasing ALBI grade and the incidence and severity of PHLF. Andreatos et al. reported a risk of PHLF of 4% for grade 1, 7.2% for grade 2 and 10% for grade 3 [26]. Therefore, individuals with ALBI grade 1 have a lower likelihood of developing PHLF and are good candidates for liver resection, while patients with ALBI grade 2 are heterogeneous and generally have a higher risk of PHLF following major hepatectomy. Finally, grade 3 contraindicates hepatic resection. Many previous studies have suggested that ALBI score is an effective predictor of PHLF and reported various cut-off values for the ALBI score in predicting PHLF after liver resection for HCC. Notably, Zou and colleagues reported a cut-off value of −2.303, which demonstrated a sensitivity of 77.3% and a specificity of 64.0% in predicting PHLF [27].

Despite ALBI score is based just on two serological markers, in the literature, it is reported that the discriminative power of the ALBI score for predicting PHLF is more accurate than the CP score [28,29]. Furthermore, ALBI score allows a better stratification of patients with CP grade A, who are usually considered good candidates for liver resection, into two categories; indeed, in this situation patients with ALBI grade 1 have a lower incidence and severity of PHLF compared to those with grade 2. This highlights the significant variability in liver functional reserve among patients with CP grade A, which can be more accurately assessed using the ALBI score in order to select patients for surgery [30]. Finally, compared to the MELD score, the ALBI score has been shown to be a better predictor of PHLF and survival outcomes in patients undergoing liver resection for HCC [31].

Although the ALBI grade is considered superior to the Child–Pugh score and the MELD score in the assessment of liver function, it has some drawbacks. For example, the ALBI grade 2 group, which represent approximately 52–65% of all HCC patients, is a wide and heterogeneous population that includes individuals with both good and poor prognosis [32]. For this reason, Hiraoka et al. proposed a modified ALBI score, which allows a subdivision of group 2 into two different prognostic subgroups: 2a (<−2.270) and 2b (≥−2.270) [33]. Kudo reported a significant disparity in median overall survival between the two subgroups for patients with HCC who underwent resection (90 months for 2a versus 62 months for 2b), and suggested that mALBI grade may be a more useful tool than ALBI score in selecting patients according to hepatic functional reserve before any treatment [32].

Another problem related to the ALBI score is the complexity of the formula used to calculate it, which may limit its use in clinical practice. To overcome this problem Kariyama and colleagues developed a simpler version called Easy-ALBI score (EZ-ALBI), which can be calculated at the bedside without requiring any technical devices. The EZ-ALBI score was created by simplifying the coefficient present in the ALBI score formula, resulting in the formula = T.Bil (mg/dL) − (9 × Alb (g/dL)) and allows to distinguish between grade 1 (score < 34.4), grade 2 (score between 34.4 to 22.2) and grade 3 (score > 22.2) [34]. The EZ-ALBI score shows a strong correlation with the original ALBI score and can be used to evaluate liver functional reserve. It also correlates well with long-term survival in patients with HCC, wherein the risk of mortality increases with the grade (2.3 times higher for grade 3 than for grade 1) [35,36,37].

### 3.4. PALBI Score

Recently, Roayaie et al. proposed to modify the ALBI score by including platelet count as a surrogate indicator of portal hypertension, resulting in the platelet–albumin–bilirubin (PALBI) grade [38]. Indeed, one of the limitations of ALBI score is that does not consider portal hypertension, which is closely related with the postoperative complication rate and PHLF in HCC patients [39]. The PALBI grade, like the ALBI score, is an objective score calculated using a mathematical formula (Table 3) that classifies patients into one of three categories: grade 1 (≤−2.53), grade 2 (−2.52 to −2.09) and grade 3 (>−2.09). A study of 2038 HCC patients with CP grade A liver function found that PALBI grade was a simple and effective method for risk stratification of PHLF and reported that the incidence of grade B/C PHLF (as defined by ISGLS) increased with increasing PALBI score, with rates of 4.4% for grade 1, 17.1% for grade 2 and 20.0% for grade 3. Similar to the ALBI score, the PALBI score can help to further differentiate CP grade A into three distinct prognostic groups [40].

### 3.5. APRI Score

The aspartate aminotransferase-to-platelet ratio index (APRI) was initially reported by Wai et al. in 2003 as a non-invasive marker for assessing hepatic fibrosis in patients with chronic hepatitis C virus (HCV) infection. This score is calculated using only two routine and inexpensive laboratory tests (AST level and platelet count) and it is determined using the formula: APRI score = (AST/upper limit of normal)/platelet count (10^9^/L) × 100 [41]. Generally, an APRI score > 1.5 indicates significant fibrosis [42].

The ability to predict the presence of fibrosis or cirrhosis in HCC patients is of utmost importance, given that research indicates a higher risk of post-hepatectomy liver failure (PHLF) and mortality following resection for HCC in individuals with hepatic fibrosis [43]. Several studies conducted in Asia demonstrated the utility of APRI score in predicting postoperative outcomes after liver resection for HCC [44,45]. The study by Rong-Yun Mai et al. revealed that patients with an APRI score above 0.55 had a higher risk of developing PHLF than those with a score below 0.55 and reported a sensitivity of 72.0% and specificity of 68% in predicting PHLF in HCC patients. Furthermore, like other authors, they suggest that the APRI score may be a more accurate predictor of PHLF than other commonly used liver function tests, such as the Child–Pugh score and MELD score [46,47]. Recently, some authors have suggested combining the ALBI and APRI score to improve their ability to predict PHLF. Although the ALBI score can assess liver function, it does not indicate the presence or absence of liver fibrosis, which can be very important since some patients may have good liver function despite severe fibrosis that increases the risk of PHLF. Liver function reserve and cirrhosis severity must be considered simultaneously in predicting PHLF and this can be achieved by combining the ALBI score and the APRI score into the ALBI-APRI score [29,48,49].

### 3.6. FIB-4 Index

The four-factor fibrosis index (FIB-4), like the APRI score, is a non-invasive tool based on a mathematical formula (Table 3) to assess liver fibrosis and was originally developed for patients with HIV/HCV co-infection. It is calculated using age, AST, ALT and platelet count, and has been shown to have a high negative predictive value (90%) for excluding advanced fibrosis when the score is below 1.45, and a moderate positive predictive value (65%) for advanced fibrosis when the score is above 3.25 [50]. Recent studies have suggested that the FIB-4 index can also be used as an independent predictor of prognosis and post-operative outcomes in patients with HCC treated with hepatectomy [51,52]. Zhou et al. recently investigated the potential relationship between FIB-4 and PHLF in patients with HCC. Their results suggest that there is a positive correlation between FIB-4 and the incidence of PHLF, with higher FIB-4 scores associated with an increased risk of PHLF. The optimal cut-off value for predicting PHLF was found to be FIB-4 > 4.16. The study authors suggest that FIB-4 may allow a better stratification of patients with CP grade A by differentiating between those with better liver functional reserve (FIB-4 ≤ 4.16) and those with worse liver functional reserve (FIB-4 > 4.16) [53].

**Table 3 medicina-59-01099-t003:** Non-invasive blood test score formulas.

Blood Test Scores	Formula
MELD [21]	9.57 × log_e_ (creatinine [mg/dL]) + 3.78 × log_e_ (total bilirubin [mg/dL]) + 11.2 × log_e_ (INR) + 6.43
ALBI [17]	0.66 × log_10_ (total bilirubin [μmol/L]) − 0.085 × (albumin [g/L])
EZ-ALBI [34]	Total bilirubin [mg/dL] − (9 × Albumin [g/L])
PALBI [38]	2.02 × log_10_ total bilirubin − 0.37 × (log_10_ bilirubin)^2^ − 0.04 albumin − 3.48 log_10_ platelets + 1.01 (log_10_ platelets)^2^
APRI [41]	(AST/upper limit of normal)/platelet count (10^9^/L) × 100
FIB-4 INDEX [51]	Age (years) × AST (U/L)/[Platelets (10^9^/L) × ALT ^1/2^ (U/L)]

Abbreviations: MELD (Model for End Stage Liver Disease); ALBI (Albumin–bilirubin index); EZ-ALBI (Easy-ALBI); PALBI (platelet-ALBI); APRI (aspartate aminotransferase-to-platelet ratio index); FIB-4 index (four-factor fibrosis index).

## 4. Discussion

The prevention and management of PHLF still remains a significant challenge in HCC surgery due to its potentially lethal consequences. Accurate preoperative prediction of PHLF may guide the selection of appropriate treatment and help in the decision for preoperative treatments, such as PVE (portal vein embolization), two-stage hepatectomy or ALPPS (associating liver partition and portal vein ligation for staged hepatectomy), which may increase liver volume and reduce the risk of PHLF [54]. Various scores and blood tests can be used to predict the occurrence of PHLF preoperatively, but selecting the appropriate patients for liver resection remains a complex task. Given the absence of a specific treatment for PHLF, the best approach is prevention through a comprehensive assessment of liver function that considers various aspects of liver function (synthetic, metabolic and excretory function) as well as the future liver remnant and the presence or absence of cirrhosis and portal hypertension [5].

Recent comparative studies have shown that the ALBI, APRI and FIB-4 scores have better predictive power for PHLF compared to the widely used CP and MELD scores (Table 4). Despite its limitations, the CP score is still considered a useful tool for evaluating preoperative liver function in cirrhotic patients and is recommended by Eastern and Western guidelines [3,4,5,6,7,8,9,10,11,12,13,14,15,16,17,18]. Its use in predicting PHLF in HCC is limited because it does not appropriately stratify these patients, many of whom belong to class A, resulting in a potential underestimation of the PHLF risk. Similarly, the MELD score may be a suitable tool in cirrhotic patients with HCC but is less useful in non-cirrhotic patients. Therefore, the CP score and the MELD score for estimating liver reserve in non-cirrhotic HCC patients may need to be combined with different scores.

In recent years, several laboratory test-based scores have been developed to better stratify patients into different liver function classes and ensure optimal treatment. One example is the albumin–bilirubin (ALBI) score, introduced in 2015 by Johnson, which is an objective measure based on serum albumin and serum bilirubin levels. However, it requires a complex mathematical formula or a calculator for computation and does not consider aspects related to portal hypertension or liver fibrosis, which are risk factors for PHLF [17]. To overcome this limitation, the Platelet-ALBI (PALBI) score was introduced, incorporating platelet count as a surrogate for portal hypertension [38].

To improve the ability of the ALBI score to predict PHLF, some researchers have suggested combining it with other quantitative or dynamic tests. One approach is to combine it with the APRI score, a non-invasive marker of fibrosis, resulting in the ALBI-APRI score [29]. This one has the advantage of simultaneously assessing liver function and the severity of cirrhosis. Another strategy is to incorporate dynamic tests such as the 15-min indocyanine green retention rate (ICGr15) into the ALBI score, resulting in a new score called the Albumin–Indocyanine Green Evaluation (ALICE) [56]. This approach showed a high predictive value for PHLF. In addition, a group of researchers led by Zhang proposed combining the ALBI score with spleen thickness (ST), a non-invasive indicator of portal hypertension that can be easily measured by abdominal ultrasound, which showed high predictive power for PHLF [55].

Non-invasive markers of fibrosis, such as FIB-4 and APRI, are also available for predicting PHLF. They focus mainly on cirrhosis-related aspects, but may be incomplete when used alone to assess HCC patients who may have isolated tumor without cirrhosis. It is important to note that the variables used in the FIB-4 and APRI scores reflect liver damage and portal hypertension and do not overlap with those used in the ALBI, MELD and CP scores. Therefore, a combination of these scores, may provide a more complete assessment of both liver function and fibrosis in patients with HCC.

In addition to these scores, liver stiffness measured by ultrasound (US) is a well-known non-invasive method for evaluating liver fibrosis, which can also assess liver function. Combining this method with blood test scores has been proposed to better stratify patients according to their risk of PHLF. For instance, Huang et al. found that liver stiffness could better stratify patients with CP class A, which is usually a low-risk group for PHLF. Patients with liver stiffness greater than 8.05 KPa required a higher liver volume remnant, indicating a higher risk of PHLF [57].

Finally, it is important to assess future liver volume (FLV) before performing liver resection and combining it with quantitative tests can improve the prediction of PHLF [58,59]. Thus, it has been proposed to combine FLV with FIB-4 to create a new score (FLR Volume Ratios/FIB-4) that can predict the probability of PHLF [60].

## 5. Conclusions

In conclusion, the prediction of PHLF remains a challenging task due to complex nature of liver function and a single comprehensive score of liver function does not exist. This review underscores the importance of combining different scores based on blood tests to improve the accuracy of PHLF prediction in patients with HCC. Blood test scores are crucial, but may not be sufficient to accurately predict PHLF. Therefore, the addition of dynamic and imaging tests, such as liver volumetry and ICG-15, to quantitative blood tests could potentially improve the predictive ability of these models.

## Figures and Tables

**Table 1 medicina-59-01099-t001:** Common definitions for PHLF.

Definition	Authors	Criteria
Postoperative Liver Failure	Eguchi et al./Dig. Dis. Sci. 2000 [7]	Hepatic encephalopathy, progressive hyperbilirubinemia and reduction in the hepaplastin test (<60% of control)
50–50 criteria	Balzan et al./Ann. Surg. 2005 [8]	Prothrombine time < 50% of normal (INR > 1.7) and serum bilirubin > 50 μmol/L (>2.9 mg/dL) on or after POD 5.
Bilirubin peak	Mullen et al./J. Am. Coll. Surg. 2007 [9]	Postoperative peak serum bilirubin > 120 μmol/mL (>7 mg/dL).
PHLF	(ISGLS) Rahbari et al./Surgery 2011 [10]	Postoperatively acquired deterioration in the ability of the liver to maintain its synthetic, excretory and detoxifying functions, which are characterized by an increased INR and concomitant hyperbilirubinemia on or after POD 5.

Abbreviations: POD (postoperative day); INR (international normalized ratio); ISGLS (International Study Group of Liver Surgery); PHLF (Post hepatectomy liver failure).

**Table 2 medicina-59-01099-t002:** The Child–Pugh scoring system.

Parameter	Numerical Score		
	1	2	3
Ascites	none	slight	moderate to severe
Encephalopathy	none	grade 1–2	grade 3–4
Bilirubin (mg/dL)	<2 mg/dL	2–3 mg/dL	>3 mg/dL
Albumin (mg/dL)	>3.5 mg/dL	2.8–3.5 mg/dL	<2.8 mg/dL
INR	<1.7	1.7–2.2	>2.2

Child A 5–6 pt; Child B 7–9 pt; Child C 10–15 pt. Abbreviations: INR (international Normalized Ratio).

**Table 4 medicina-59-01099-t004:** Comparison of comparative studies regarding predictive ability for PHLF.

Author	Comparative Analysis	Type of Study	Population		Conclusion
Y.-Y. Wang/2015 [30]	ALBI vs. CP	retrospective	HCC	1242	ALBI score is superior to CP score
Pan Zhou/2019 [53]	FIB-4 vs. CP	retrospective	HCC	495	FIB-4 is superior to CP score
Heng Zou/2018 [27]	ALBI vs. CP, MELD and ICGr15	retrospective	HCC	473	ALBI score is better than the Child–Pugh score, the MELD score and the ICG R15.
Ze-Qun Zhang/2019 [55]	ALBI-ST vs. APRI and FIB-4	retrospective	HCC	932	ALBI/ST ratio is better than the APRI and the FIB-4
Rong-yun Mai/2019 [29]	ALBI, APRI and ALBI-APRI vs. CP	retrospective	HCC	1055	ALBI and APRI scores are better than C-P and ALBI-APRI is superior compared to either score alone.
Jin-Yu Shi/2021 [47]	ALBI, APRI vs. CP and MELD	retrospective	HCC	767	APRI and ALBI are better than CP and MELD scores

Abbreviations: ALBI (albumin–bilirubin index); ALBI-st (ALBI–spleen thickness); CP (Child–Pugh); FIB-4 (Four Factor fibrosis index); MELD (Model for End Stage Liver Disease); APRI (aspartate aminotransferase-to-platelet ratio index).

## Data Availability

The authors confirm that the data supporting the findings of this study are available within the article.

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
