# Peer review of "Predicting Post-Hepatectomy Liver Failure in HCC Patients: A Review of Liver Function Assessment Based on Laboratory Tests Scores"

_medicina, 2023, doi:10.3390/medicina59061099_

Round 1

Reviewer 1 Report

In the presented article, the authors discussed ,,Predicting post-hepatectomy liver failure in HCC patients: a review of liver function assessment based on laboratory test scores’’.

Research is interesting and has an application function. The authors showed that the best way to assess the risk of post-hepatectomy liver failure is to combine several methods (blood tests, imaging tests) rather than relying on only one.

The authors used the current literature. The summary work is very good. The results were broadly discussed. This work has very practical applications. 

Author Response

We really appreciate your feedback on our article and we are glad for your positive comments on our research. It is nice to know that you found our manuscript interesting and practical. Thank you for taking the time to review our article and we hope that our research will be useful to clinicians in this field.

Sincerely,

Alessio Morandi

Reviewer 2 Report

Assessment of liver function is crucial in predicting the risk of post-hepatectomy liver failure (PHLF) in patients undergoing liver resection, especially in patients with hepatocellular carcinoma (HCC) who are often associated with liver cirrhosis. In this manuscript, the authors reviewed the blood tests and scores for assessing hepatic function, including the Child-Pugh score (CP score), model for end stage liver disease (MELD) score, albumin-bilirubin score (ALBI) and related scores (allowing the most accurate prediction of PHLF for HCC patients), platelet-albumin-bilirubin (PALBI) grade, FIB-4, APRI, and the combination of ALBI-APRI score. The authors concluded that blood test scores may be combined to achieve a better predictive value of PHLF, and suggested that the inclusion of dynamic and imaging tests such as liver volumetry and ICG r15 may be helpful to improve the predictive capacity of these models.

This is a comprehensive review on the blood tests and scores for assessing hepatic function and predicting the risk of post-hepatectomy liver failure. The manuscript was well prepared. This review provided useful information for the clinicians to manage the patients undergoing liver resection. 

Author Response

Thank you for the positive review of our manuscript. We are glad that you found our review useful for clinicians. Our goal was to provide information on the most common blood test scores and to provide a tool to guide clinicians in selecting HCC patients for liver resection. Once again, thank you for your review and feedback on our manuscript.

Sincerely,

Alessio Morandi

Reviewer 3 Report

In the manuscript entitled “Predicting of post-hepatectomy liver failure in HCC patients: A review of Liver function assessment based on laboratory test scores” Morandi et al., have investigated prediction of post-hepatectomy liver failure (PHLF) in HCC patients. So far various methodology (Invasive and Non-invasive) has been proposed to score the perdition of liver failure in HCC patients but still, they are not satisfactory and accurate enough because of many clinical limitations and challenges. In this manuscript, they have done an extensive literature survey and proposed a non-invasive, least expensive, blood-based characterization of all relevant associated markers of liver failure in HCC patients so that a combined score may be more reliable and sensitive for clinical settings. The manuscript is well-written and nicely laid out so that all readers can have a better understanding of the subject.

It is recommended that to publish this work following minor comments can be responded, and suggestions can be addressed in the revised manuscript.

1.     This paper is interesting (See the link) and may be very helpful to justify why we need an innovative combined predictive score, would have been better to cite it in the manuscript.

 https://pubmed.ncbi.nlm.nih.gov/36977863/

 2.     What is the drawback of spleen-volume-to platelet ratio, not mentioned in the manuscript? Can be highlighted in the discussion section or some other relevant sections.

https://www.sciencedirect.com/science/article/pii/S1015958422005450

Author Response

Thank you for taking the time to review my manuscript and for your feedback. We really appreciate your valuable comments. We are glad to know that you found our manuscript interesting and well written. We have carefully considered your suggestions and partially incorporated them into the revision of the manuscript.

Regarding the first comment, we read the article entitled "Predicting Post hepatectomy Liver Failure Preoperatively for Child Pugh A5 Hepatocellular Carcinoma Patients by Liver Stiffness" and found it very informative and relevant to the topic of our manuscript. we have therefore added this article to the discussion section.

Regarding the second article you mentioned (A nomogram for predicting post-hepatectomy liver failure in patients with hepatocellular carcinoma based on spleen-volume-to-platelet ratio),  we appreciate your suggestion, but we did not find it directly relevant to our manuscript. The article discusses the spleen-volume-to-platelet ratio as part of a normogram (including BMI, age, blood loss, total bilirubin, ascites and prealbumin) with the aim of assessing the risk of PHLF. Given our manuscript focuses on laboratory test scores for assessing liver function and risk of PHLF, we did not find it appropriate to include this article in our discussion.

Thank you again for your valuable feedback. Please let me know if you have any further suggestions or comments.

Sincerely,

Alessio Morandi
